# Finite Element Analysis of a New Non-Engaging Abutment System for Three-Unit Implant-Supported Fixed Dental Prostheses

**DOI:** 10.3390/bioengineering9100483

**Published:** 2022-09-20

**Authors:** Soo-Hwan Byun, Joung-Hwa Seo, Ran-Yeong Cho, Sang-Min Yi, Lee-Kyong Kim, Hyun-Sook Han, Sung-Woon On, Won-Hyeon Kim, Hyun-Wook An, Byoung-Eun Yang

**Affiliations:** 1Department of Oral and Maxillofacial Surgery, Hallym University Sacred Heart Hospital, Anyang 14068, Korea; 2Graduate School of Clinical Dentistry, Hallym University, Chuncheon 24252, Korea; 3Institute of Clinical Dentistry, Hallym University, Chuncheon 24252, Korea; 4Department of Prosthodontics, Hallym University Sacred Heart Hospital, Anyang 14068, Korea; 5Division of Oral and Maxillofacial Surgery, Department of Dentistry, Hallym University Dongtan Sacred Heart Hospital, Hwaseong 18450, Korea; 6Dental Life Science Research Institute/Innovative Research & Support Center for Dental Science, Seoul National University Dental Hospital, Seoul 03080, Korea; 7Research and Development Center, MegaGen Implant Co., Ltd., Daegu 42921, Korea

**Keywords:** dental implant–abutment design, finite element analysis, biomechanics, implant-supported, dental prostheses

## Abstract

(1) Background: The stability of implants plays a significant role in the success of osseointegration. The stability of the connection between the fixture and the abutment is one of the critical factors affecting osseointegration. When restoring multiple, non-parallel, and splinted implants, achieving a passive fit can be complicated and challenging. A new EZ post non-engaging abutment system of the BlueDiamond^®^ (BD) implant allows a wide connection angle while achieving a passive prosthesis fit. This study aimed to confirm the new abutment system’s clinical applicability by evaluating its biomechanical characteristics using finite element analysis (FEA). (2) Methods: The implant-supported fixed three-unit dental prostheses model was reproduced for two groups of AnyOne^®^ (AO) and BD implants using FEA. The loading conditions were a preload of 200 N in the first step and loads of 100 N (axial), 100 N (15°), or 30 N (45°) in the second step. (3) Results: The peak Von Mises stress (PVMS) value of the fixture in the BD group was more than twice that in the AO group. In contrast, the PVMS values of the abutment and abutment screws were lower in the BD group than in the AO group. The AO group revealed higher maximal principal stress (MPS) values than that of the BD group in the cortical bone, cancellous bone, and crown. The average stress of the outer surface of the abutment was lower in the AO group than in the BD group. The stress distribution for the inner surface of the fixture confirmed that the BD group displayed a lower stress distribution than the AO group under axial and 15° loads; however, the average stress was 1.5 times higher at the 45° load. The stress values of the entire surface where the cortical and cancellous bone were in contact with the fixture were measured. The AO group showed a higher stress value than the BD group in both cortical and cancellous bone. (4) Conclusions: In the AO group, the PVMS value of the fixture and the stress distribution at the contact surface between the fixture and the abutment were lower than those of the BD group, suggesting that the stability of the fixture would be high. However, due to the high stress in the fastening area of the abutment and abutment screw, the risk of abutment fracture in the AO group is higher than that of the BD group. Therefore, the new EZ post non-engaging abutment of the BD implant can be used without any problems in clinics, similar to the non-engaging abutment of the AO implant, which has been widely used in clinical practice.

## 1. Introduction

Implant-supported fixed dental prostheses (ISFDPs) are a standard treatment option for partially or completely edentulous individuals. ISFDPs can be used to restore multiple consecutive crowns while reducing the number of fixtures that must be placed [1,2]. However, when restoring ISFDPs, appropriate occlusal load distribution is essential for long-term success. Rangert and Jemt recommended that adjacent implants should be rigidly connected using fixed screw-retained restorations to distribute the occlusal load better and minimize screw loosening [3]. De Souza Batista et al. compared the survival rate of splinted and non-splinted adjacent implants and observed that implant-supporting splinted restorations showed statistically significantly higher survival rates [4].

However, achieving a passive fit can be complex and challenging when restoring multiple, non-parallel, and splinted implants [5]. For bone-level implants with conical (or tapered) internal connections, the placement angle between fixtures affects the prosthesis’ insertion path. When the placement angles between fixtures are not parallel, splinted cement-retained restorations are fabricated and then bonded using temporary cement. However, this connection method can leave dental cement below the gingiva. Consequently, the bonding force between the prosthesis and the abutment or separation of the prosthesis from the abutment is difficult to control when screw loosening occurs. The frequently used internal conical connection implant system, internal indexed fixtures, and engaging abutments enhance the stability of the prosthesis by allowing the abutment’s connection to the exact location. However, the insertion or removal of non-parallel ISFDPs can be challenging [2,6].

The most frequently used type is a hex, anti-rotational abutment with 22°-tapered internal fixture [7]. When two or more implants are placed, there is a high possibility of prostheses misconnection if the three-dimensionally misaligned angle exceeds 22° [8]. Therefore, a dental implant company recommends the use of convertible abutment if the path is off by >22° [9]. This phenomenon cannot be easily distinguished by cone-beam computed tomography (CBCT) or radiography, and when an incorrect tightening is made, constrained tightening may occur due to the tightening force of the abutment screw [10]. In such cases, stress concentration of the masticatory force occurs due to the misaligned angle, leading to prosthesis, abutment, or fixture fracture [11].

Thus, there is an emerging need for an abutment for ISFDPs that allows a certain degree of correction.

A multi-unit abutment can solve this problem. However, the screw holding the fixed dental prostheses is relatively tiny, making it clinically challenging to manipulate the screw. Moreover, additional fabricated caps may be required to obtain a better parallel path [12], and the use of an additional multi-unit abutment increases the treatment cost.

Non-engaging abutments (abutment without index) are another option that allow a certain degree of correction, achieving an excellent prosthesis passive fit to multiple, non-parallel internally indexed implants [2]. Non-engaging abutments are mostly used for multi-unit cases where they need to have a passive fit and not put too much force on the implants [13].

Engaging abutments are made to lock into the interface of the fixture’s unique anti-rotational shape (hex, star, square, octa, etc.). However, this anti-rotational property is absent from a non-engaging abutment. Implant restorations using two or more units can be placed side by side to prevent individual abutments from rotating on the supporting fixtures. Thus, the inclusion of built-in optional anti-rotational features is theoretically unnecessary. In these situations, non-engaging abutments are used for splinted screw-retained restorations.

Considering the necessity of an implant–abutment system with a wide connection angle, and the mechanical advantages of the splinted crowns, the new EZ post non-engaging abutment system for BlueDiamond^®^ (BD, MegaGen, Daegu, Korea) fixtures, which have a 30° connection angle, has been devised for screw-cemented retained ISFDPs [14]. The BD fixture has an arch-type keystone structure that can minimize the rotation angle for a precise connection with the abutment. In ISFDPs, this keystone part of the abutment must be removed to secure the connection angle. Therefore, the connecting length of the abutment inserted into the fixture is reduced by eliminating the abutment index structure. The thickness of the abutment screw for non-engaging abutment was increased to compensate for the stress concentration on the abutment screw caused by changes in the structure. Accurate tightening of the prosthesis can prevent unfavorable complications such as fractures in the implant system, pain, marginal bone loss, and even loss of osseointegration [15].

A non-hex multiple ISFDP abutment has been extensively used in clinical practice. However, studies on the stress distribution of new non-engaging abutment systems are insufficient. Therefore, in this study, the AnyOne^®^ (AO; MegaGen, Daegu, Korea) implant system (Figure 1A) with a non-engaging (non-hex) abutment with a 22° connection angle fixture was used in the control group (AO group) (Figure 2A), and the BD^®^ implant system with a non-engaging (non-octa) abutment was used in the test group (BD group) (Figure 2B). According to Lin et al., finite element analysis (FEA) accurately measures the mechanical response of fixed dental prostheses [16]. The widespread use of FEA is due to its low cost, predictability, and steadily increasing accuracy. Furthermore, FEA can help study how an implant, prosthesis, and bone complex react to mechanical loading [2]. An implant-supported fixed three-unit dental prostheses model was reproduced for the AO and BD groups through FEA, and a comparative analysis was performed in three occlusions. The purpose of this study was to test the null hypothesis of no difference in the stress distribution of non-engaging abutments between the AO and BD groups.

## 2. Materials and Methods

### 2.1. Design and Dimensions of the Fixture, Abutment, and Abutment Screw

The fixture design in the AO group had a diameter of 4.5 mm, length of 8 mm, and screw thread pitch of 0.8 mm. A profile of 5.5 mm, cuff height of 3.5 mm, and post height of 5.5 mm was used for the abutment in the second premolar. Furthermore, a profile of 6.5 mm, cuff height of 3.5 mm, and post height of 5.5 mm was used for the abutment utilized in the second molar. The abutment screw had a head diameter of 2.3 mm, a body diameter of 1.96 mm, a length of 10.1 mm, and a screw thread pitch of 0.4 mm. The fixture design in the BD group had a diameter of 4.4 mm, length of 8 mm, and screw thread pitch of 0.8 mm. In addition, for the abutment used in the second premolar, the profile was 5 mm, the cuff height was 3 mm, and the post height was 5.5 mm. For the abutment used in the second molar, the profile was 6 mm, the cuff height was 3 mm, and the post height was 5.5 mm. The abutment screw had a head diameter of 2.6 mm, a body diameter of 2.38 mm, a length of 9.9 mm, and a screw thread pitch of 0.35 mm (Figure 1 and Figure 2).

### 2.2. Bone Shape and Composition of Surgical Model

#### 2.2.1. Finite Element Bone Shape Model

We utilized a three-dimensional (3D) mandibular FEA model that was previously studied (Figure 3) [17] to compare the stress distribution inside the bones of both groups. A two-dimensional image of a standard Korean male adult skull from a CBCT scan was cut into 0.25 mm-thick pieces, and the pieces were then used to reconstruct three-dimensional mandible models. The cortical bone was fixed to a constant thickness of 2 mm, and the cancellous and cortical bones were identified [18].

#### 2.2.2. The Surgical Model Composition

A surgical model was reproduced for two-implant systems. An implant system was established for the surgical model in the second premolars and second molars, and the angle between the fixtures was 20°. An implant-supported three-unit fixed dental prostheses model was constructed by extracting the teeth shape corresponding to the second premolar, first molar, and second molar from the CBCT images. The same crown model was used for comparatively analyzing the designs in the AO and BD groups, and the depth of fixture placement was matched to the uppermost position of the fixture (Figure 4).

### 2.3. Material Properties

The properties of the components used in this study were based on those used in the previous literature and are listed in Table 1 [19,20,21,22] and Table 2.

### 2.4. Loads, Boundaries, and Contact Conditions

Since the mandible does not move during occlusion, both sides of the cortical and cancellous bones were completely restrained to prevent translation and rotation in the x, y, and z directions, as shown in Figure 5 [17]. No restraint conditions were applied to dental implant systems (abutment, fixture, and abutment screw) and crowns except for cortical and cancellous bones. As for the bonding state between the bone and the implant system, “tie contact” was applied, assuming that the fixture was placed and fused into the bone. Moreover, considering the state in which the cortical and cancellous bones are completely combined, “tie contact” was applied. Likewise, “tie contact” was applied because the crown and abutment are completely connected by adhesive. A friction coefficient of 0.5 and a “surface-to-surface” contact condition were applied for the connection status between the abutment–fixture, abutment screw–fixture, and abutment–abutment screw [20]. All structures to which “tie contact” and “surface-to-surface” conditions were applied were allowed to move and rotate in all directions.

The FEA was performed in two stages. In the first stage, a preload of 200 N of the abutment screw was applied in the vertical direction to consider the complete fastening of the abutment and fixture [20,23]. It has been reported that the oblique load affects bone remodeling at the bone–implant interface [24], and it has been reported in the literature that the range of physiological loads in the horizontal and vertical directions is 30–100 N [25]. Therefore, in this study, the load direction for the vertical and horizontal directions was also considered regarding these studies. After the abutment and fixture were completely fastened, loads of 100 N [26], 100 N at 15 degrees [27], and 30 N at 45 degrees [26] were applied vertically to compare and analyze the stress distribution generated on the implant by occlusion (Figure 5).

### 2.5. Variables

The maximal principal stress (MPS) values were derived from the crown, cortical bone, and cancellous bone. The peak von Mises stress (PVMS) values were derived from the fixture, abutment, and abutment screws. Finally, the average value was evaluated by deriving the maximum equivalent stress (max EQS) value for the nodes in contact with the fixture body and abutment. The average values were compared by deriving the max EQS values for all the nodes in contact with the fixture in the cortical and cancellous bone.

## 3. Results

### 3.1. Stress on the Dental Implant System

The PVMS values of the fixture, abutment, and abutment screw were compared (Figure 6 and Appendix A). The PVMS value of the fixture in the BD group was more than twice that in the AO group. In contrast, the PVMS values of the abutment and abutment screws were lower in the BD group than in the AO group. The AO group displayed a three-fold-higher PVMS value for the abutment than the BD group. Furthermore, for the abutment screw, the highest stress was generated in the fourth to fifth thread positions in the AO group, and the BD group also had the highest stress in the fourth thread region (Figure 7).

### 3.2. Stress on the Cortical Bone, Cancellous Bone, and Crown

For the cortical bone, cancellous bone, and crown, the stress value was derived as MPS rather than PVMS. The AO group presented higher stress values than the BD group in the cortical bone, cancellous bone, and crown (Figure 8 and Appendix A).

### 3.3. Stress on the Inner Surface of the Abutment and Fixture

Stress was derived from the surface in contact with the abutment and fixture since they were combined and measured in two areas of the second premolar and second molar, respectively. The average stress on the outer surface of the abutment was lower in the AO group than in the BD group under all load conditions (Figure 9 and Appendix A) (Figure 10) (Figure 11). These findings confirmed that the BD group presented a better stress distribution than the AO group under the axial and 15° load in the stress distribution for the inner surface of the fixture. However, the average stress of the BD group at the 45° load was 1.5 times higher than the AO group.

### 3.4. Stress Results in the Surface of Cortical Bone and Cancellous Bone in Contact with the Fixture

The stress values of the entire surface where the cortical and cancellous bone were in contact with the fixture were measured (Figure 12 and Appendix A) (Figure 13). The stress distribution in the cancellous bone was higher in the AO group than in the BD group in the second premolars and molars. The AO group presented approximately 1.3-fold higher stress distribution than the BD group under the vertical and 15° loads. The difference at the 45° load was smaller than that at the vertical and 15° loads. The stress distribution in the cortical bone in the second premolars and molars was higher in the AO group than in the BD group. The AO group was approximately 1.3-fold higher in vertical load and 15° load. The two groups had no significant difference at the 45° load.

## 4. Discussion

Among the investigated approaches for rehabilitating three-unit edentulous areas, the three-unit implant-supported bridge supported by two fixtures has a low prevalence of peri-implantitis, is cost-effective, and provides the ideal long-term treatment solution [28]. Therefore, we simulated a loss model of three teeth in the posterior mandible and fabricated a three-unit fixed dental prosthesis supported by two fixtures in this study.

Connecting the implant prostheses of the molars effectively disperses the occlusal force applied to them, efficiently solving the food-packing phenomenon caused by the loss of contact between the prostheses [29]. The implant prostheses are also connected if the bone quality of the area of the fixture placement is poor [30], the patient’s occlusal force is strong, or if the diameter or length of the implant is narrow or relatively short, respectively [31,32,33]. When restoring multiple prostheses, placing the implants in parallel is an effective approach for optimal stress distribution [34]. However, due to the bone anatomy, implant fixtures may be inserted at angles to each other [35]. In addition, splinted restorations exhibit better load-sharing than non-splinted restorations; therefore, restoration of multiple non-parallel splinted implants is a common clinical practice [36].

Consequently, ISFDP misconnection is frequently observed in clinical cases [37]. These microgaps due to misconnection at the implant–abutment interfaces may cause microorganisms around the fixture and generate micromovements at the threads of the bolted joint [38]. In the EZ post non-engaging abutment of BD implants, the connecting part (index part) of the abutment was removed to achieve a passive fit. However, when an eccentric force is applied, a greater stress concentration can occur around the abutment screw [2]. Therefore, the thickness of the abutment screw was increased to compensate for the abutment screw stress concentration caused by changes in the abutment structure (BD in Figure 7). In a general dental implant system, the same abutment screw is used for engaging and non-engaging abutments. However, in the BD implant non-engaging abutment, the shank of the abutment screw is thicker (abutment screw in Figure 2B), unlike the screw used in the engaging abutment. This thick screw is a cause of lower stress than the AO abutment screw under various loading conditions. According to Hansson, a larger screw diameter correlates with stronger load bending [39].

FEA has helped study stress distribution in implants and the surrounding bones for 20 years because it is suitable for analyzing geometrically complex implants and the surrounding bones [40]. For the surgical model, the angle between the fixtures was 20° to assume non-parallel-placed implants. The load transfer mechanism within the implant may vary depending on the shape of the abutment and abutment screw, affecting the stress generated in the surrounding bone [41]. This study used FEA to compare the biomechanical characteristics relative to the abutment and abutment screw design changes. Two loading steps were used in this study [42]. The first loading step applied a preload to complete the tightening of the abutment screw. Then, the occlusal force was applied to the crown. In this study, two directions of load applied to the crown were considered. It has been reported that various loading directions, such as vertical, oblique, and horizontal, are considered based on the teeth in the mastication process [25,26,27,43]. In this study, to apply the various load directions generated by masticatory movement in the oral cavity, load conditions of 100 N vertically, 100 N diagonally, and 30 N horizontally were applied to the molars, which generate the maximum load during tooth occlusion. The MPS analysis identifies the weak points and strengths due to tension. When the stress value is large, the tensile strength is weak [44]. The surrounding bone response is predicted through principal stress; therefore, the MPS values were derived from the crown, cortical bone, and cancellous bone. The PVMS value was derived from the fixture, abutment, and abutment screws to analyze fracture risk. In addition, the average value was checked by deriving the max EQS value for the nodes in contact with the fixture body and the abutment and nodes in contact with the fixture in the cortical and cancellous bone. A lower max EQS value indicates that the bone around the implant is at a lower risk of failure and has a high success rate [45].

The stress transmission from the implant system to the bone in the axial loading was better than that in the lateral loading (15° and 45°). Excess stress concentration in the peri-implant bone and deformation beyond the bone’s physiological limit cause microcracks and bone resorption [46]. Compared with the AO group, the BD group displayed less stress concentration in the bone. Therefore, the BD implant system had less effect on the bone, possibly resulting in a higher implant success rate [47].

The stress was derived from the surface in contact with the abutment and fixture since they were combined and measured in two areas of the second premolar and molar, respectively. The average stress on the outer surface of the abutment was lower in the AO group than in the BD group, and the BD group had higher stress under all load conditions. The stress distribution for the inner surface of the fixture confirmed that the BD group had a lower stress distribution than the AO group under the axial and 15° load. However, its average stress was 1.5 times higher at the 45° load. This could be attributed to the decreased extension of the abutment into the internal connection of the implant fixture. This increase in localized stress could cause material wear and loss, leading to a breakdown of the implant–prosthetic assembly [48]. Since loosening and fracture of the implant-supported prosthesis are usually associated with overloading on components, especially lateral loading, in principle, any transverse force on the prosthesis of implants should be eliminated [49,50]. When excessive lateral force is not applied to the implant-supported prosthesis according to the implant prosthetic restoration principle, the abutment and fixture of the BD group should present good clinical results.

This study has several limitations. First, dental implants must function in the body for a long time, but this study did not evaluate the type of problems in the event of bone loss due to periodontal disease. Although not an implant-related study, one study reported that the stress on the post used for root canal restoration differs according to the loss of alveolar bone [43]. Therefore, the stress on the non-engaging abutment system is expected to be different if there is alveolar bone loss. Additional studies are needed on the condition of alveolar bone loss. Second, the PVMS value of the fixture was lower in the AO group than in the BD group. One of the reasons for this difference is probably the different diameters of the fixtures in the two groups, i.e., the AO fixture is 0.1 mm thicker than the BD fixture. Since there is no fixture of the same diameter between the two implants, it was impossible to compare them under identical conditions. Third, to confirm whether the new abutment system of BD implants is clinically applicable, FEA was performed by evaluating the biomechanical characteristics. Among the various methods for applying impact to a living body and analyzing stress, the finite element method has high repeatability and allows 3D displacement and stress measurement. The design characteristics of the implant were analyzed by considering three types of static loads for the dental bridge model. The stress distribution and biomechanical characteristics of the bone and implant according to different designs were confirmed in this study. However, since the study was conducted considering only static load, a study applying fatigue load to analyze long-term durability was not conducted. Therefore, to confirm the biomechanical characteristics of these models, additional studies applying fatigue and complex loads are needed. Furthermore, preclinical and clinical studies are required due to the complexity of the implant system components and the low predictability of oral loading conditions.

## 5. Conclusions

Using the finite element method, the PVMS value of the fixture in the AO group and the stress distribution on the contact surface between the fixture and the abutment were lower than those of the BD group. However, due to the high stress in the fastening area of the abutment and abutment screw, the risk of fracture of the abutment or abutment screw is higher than that of the BD group.

Therefore, the new EZ post non-engagement abutment of the BD group can be used as well as the non-engagement abutment of the AO group, which is widely used so far.

Furthermore, this study confirmed that the BD group had a smoother stress transfer from the implant to the bone and less stress concentration in the bone than the AO group.

Therefore, when the stress around the fixture appears to be concentrated in the BD group, the stress concentration phenomenon in the cortical and cancellous bone is reduced, and the initial stability and osseointegration of the implant after surgery are expected to be good.

## Figures and Tables

**Figure 1 bioengineering-09-00483-f001:**
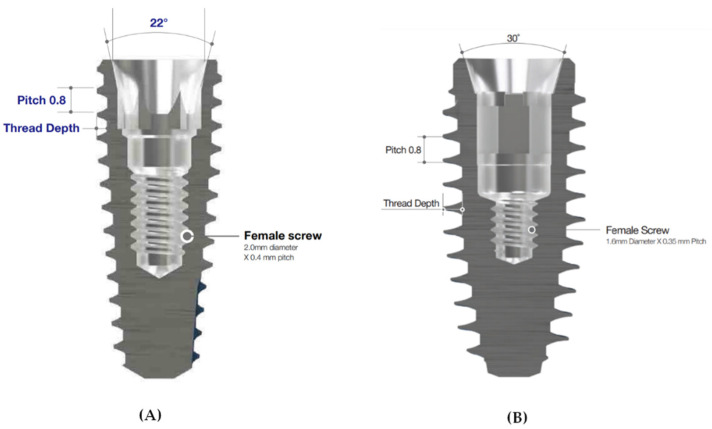
Schematic diagram of the fixtures. (**A**) AnyOne^®^ fixture with 22° connection angle; (**B**) BlueDiamond^®^ fixture with 30° connection angle.

**Figure 2 bioengineering-09-00483-f002:**
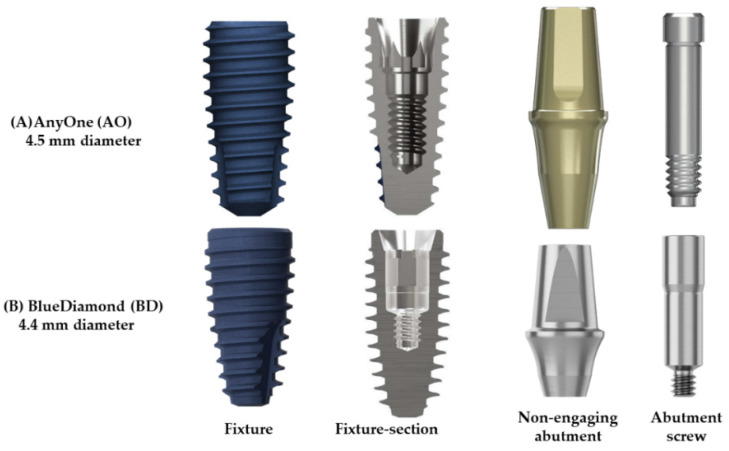
Designs and dimensions of the fixture, abutment, and abutment screw. (**A**) AnyOne^®^ fixture with 4.5 mm diameter; (**B**) BlueDiamond^®^ fixture with 4.4 mm diameter.

**Figure 3 bioengineering-09-00483-f003:**
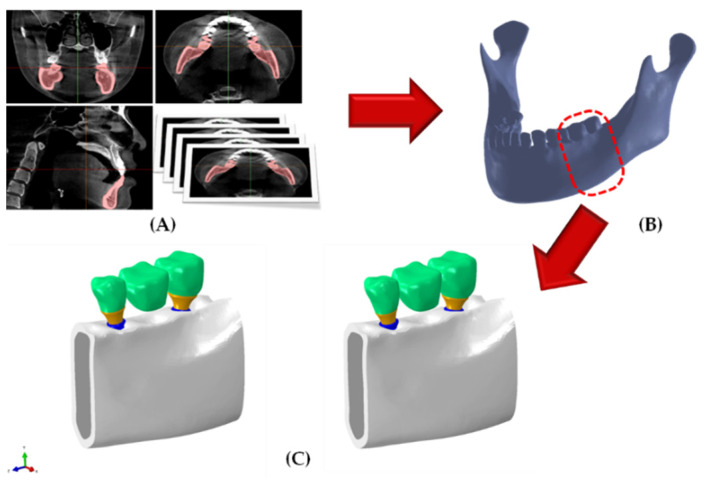
Mandible reconstruction and implant-supported fixed three-unit fixed dental prostheses. (**A**) Geometry data extraction from cone-beam computed tomography images; (**B**) reconstruction of surface and volume for mandible bone; (**C**) finite element analysis surgical model.

**Figure 4 bioengineering-09-00483-f004:**
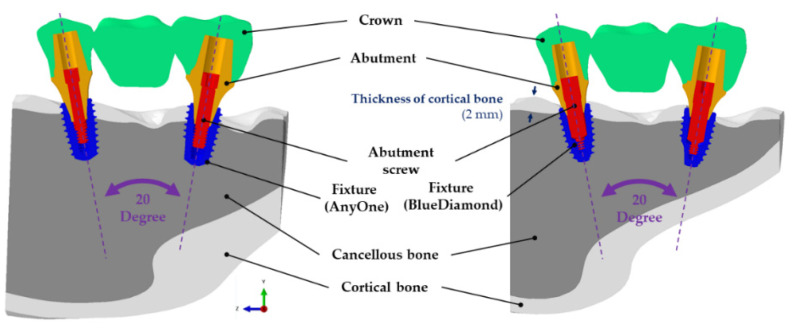
Construction of the finite element surgical model, AnyOne group and BlueDiamond group.

**Figure 5 bioengineering-09-00483-f005:**
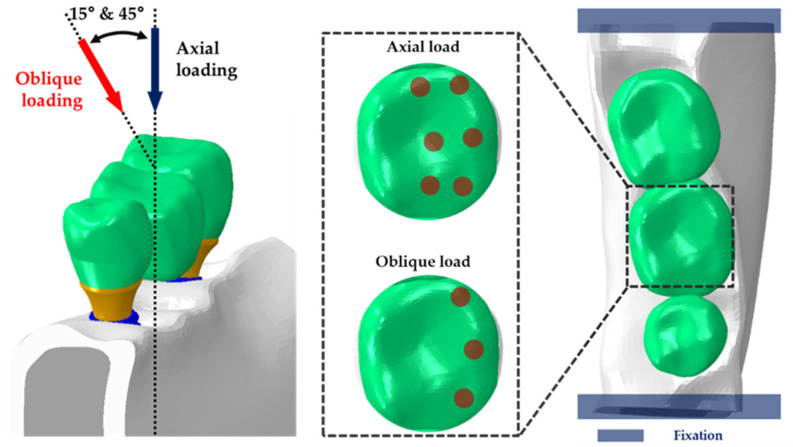
Loads and boundary conditions of the finite element model. A load of 100 N was applied on six points in the axial direction of the fixture, and loads of 100 N (15°) and 30 N (45°) were applied to three points in a non-axial direction.

**Figure 6 bioengineering-09-00483-f006:**
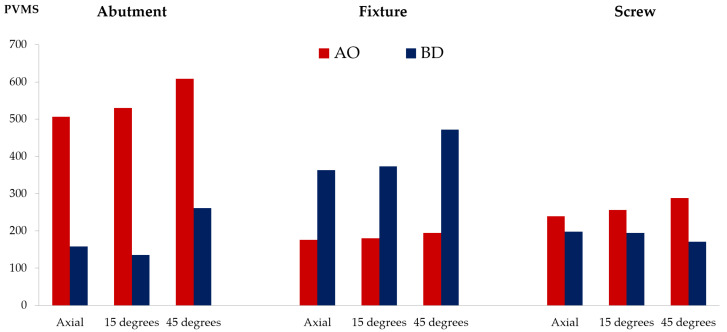
Peak von Mises stress results of the abutment, fixture and screw.

**Figure 7 bioengineering-09-00483-f007:**
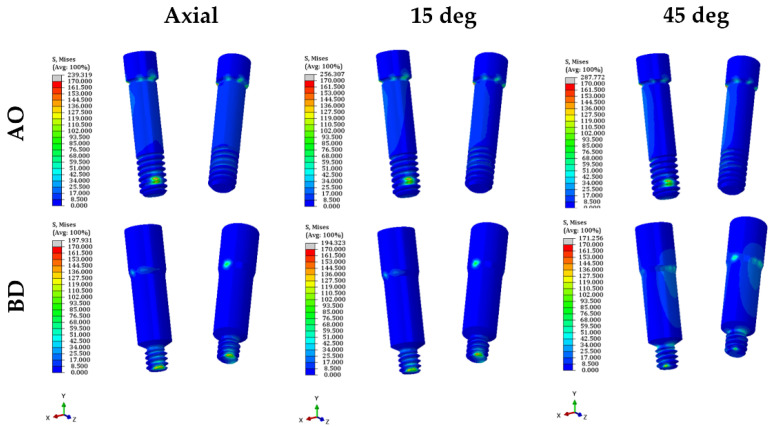
Stress distribution diagram of abutment screw in both models.

**Figure 8 bioengineering-09-00483-f008:**
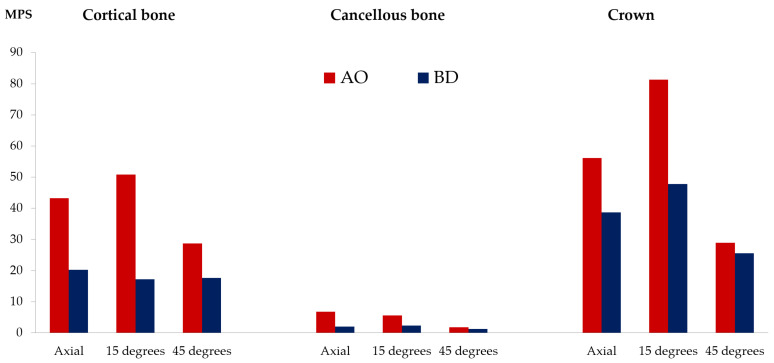
Maximal principal stress (MPa) results from cortical bone, cancellous bone, and crown.

**Figure 9 bioengineering-09-00483-f009:**
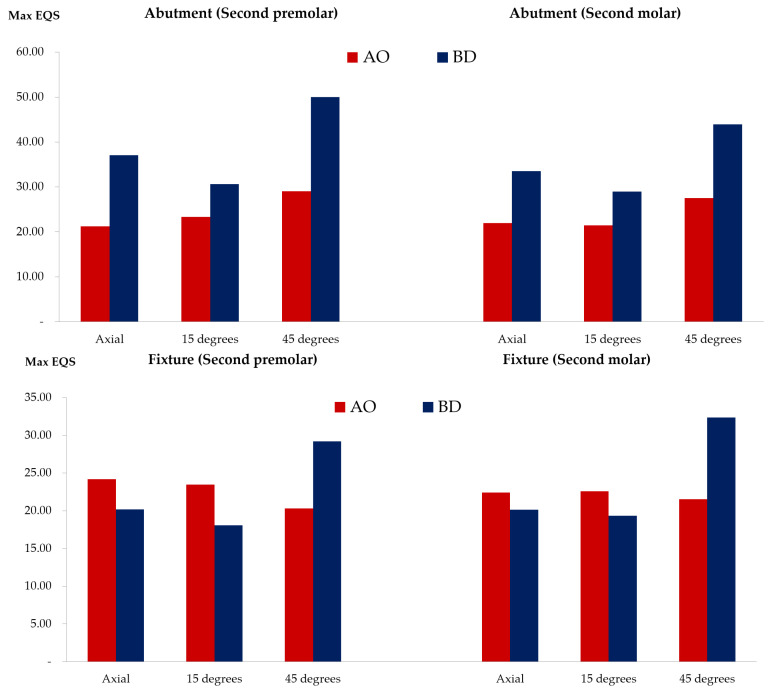
Mean stress distribution of the outer surface of the abutment and fixture in the second premolar and second molar.

**Figure 10 bioengineering-09-00483-f010:**
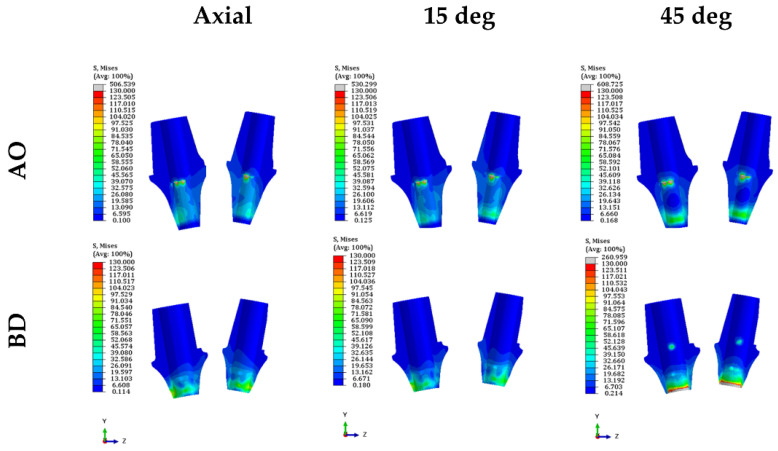
Stress distribution diagram of the abutment in both models.

**Figure 11 bioengineering-09-00483-f011:**
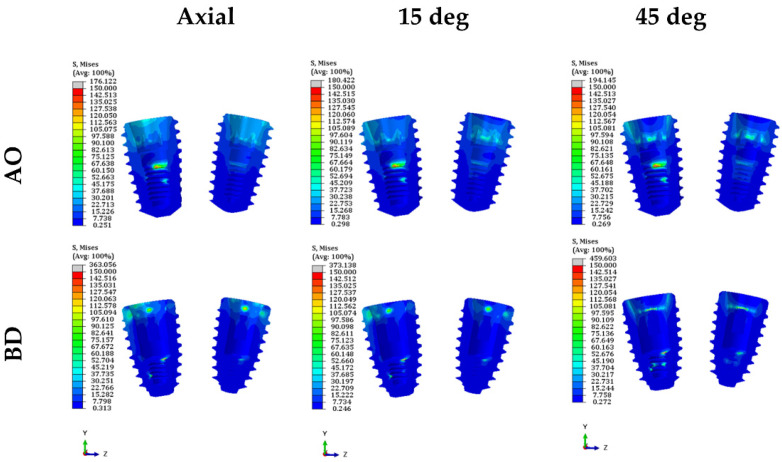
Stress distribution diagram inside the fixture of both models.

**Figure 12 bioengineering-09-00483-f012:**
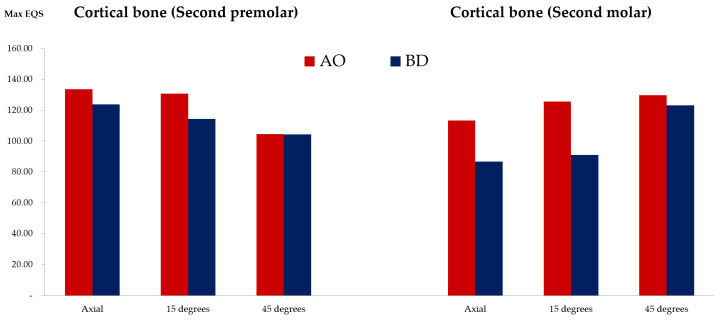
Average stress distribution from the inner surface of the cortical and cancellous bone in the second premolar and second molar areas.

**Figure 13 bioengineering-09-00483-f013:**
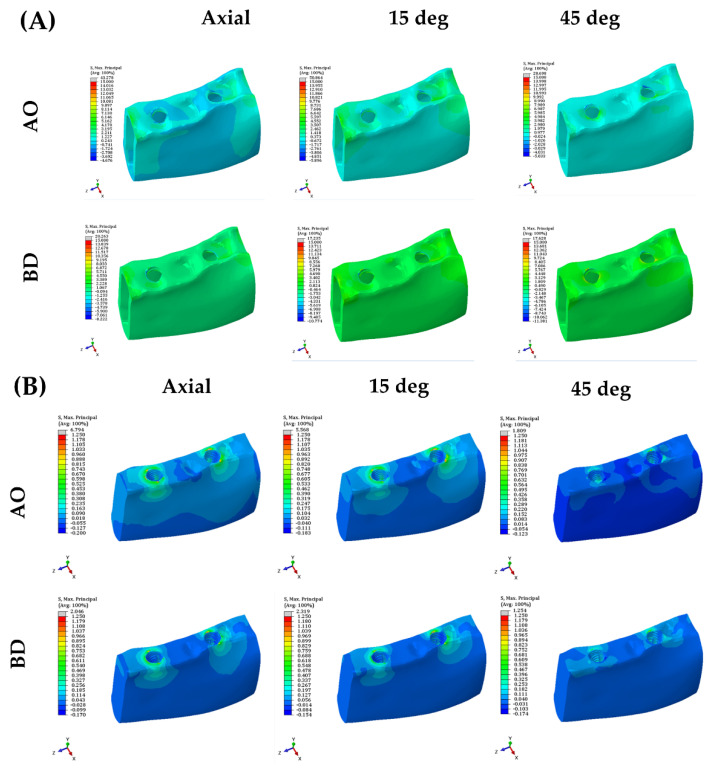
Overall stress distribution of the cortical and cancellous bones in both models. (**A**) cortical bone; (**B**) cancellous bone.

**Table 1 bioengineering-09-00483-t001:** Material properties applied to the finite element model.

Components	Young’s Modulus (MPa)	Poisson’s Ratio	Reference
Crown (zirconia)	205,000	0.19	[19]
Cortical bone	13,000	0.3	[20]
Cancellous bone	690	0.3	[20]
Abutment	114,000	0.33	[21]
Fixture	105,000	0.34	[22]
Abutment screw	114,000	0.33	[21]

**Table 2 bioengineering-09-00483-t002:** Number of elements and nodes for finite element surgical model.

Components		Second Premolar		Second Molar	
		Elements	Nodes	Elements	Nodes
AO	Cortical bone	45,667	231,127	45,667	231,127
	Cancellous bone	90,742	481,650	90,742	481,650
	Crown	35,620	166,045	35,620	166,045
	Abutment	16,589	73,483	20,050	91,142
	Fixture	24,877	109,069	24,810	108,760
	Abutment screw	14,324	72,302	14,324	72,302
BD	Cortical bone	43,103	200,130	43,103	200,130
	Cancellous bone	88,849	473,283	88,849	473,283
	Crown	43,203	222,055	43,203	222,055
	Abutment	15,861	74,119	17,622	84,378
	Fixture	40,633	209,131	40,190	206,042
	Abutment screw	17,396	90,249	17,396	90,249

## Data Availability

The datasets generated during and/or analyzed during the current study are available from the corresponding author on reasonable request.

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
