# Peer review of "Finite Element Analysis of a New Non-Engaging Abutment System for Three-Unit Implant-Supported Fixed Dental Prostheses"

_bioengineering, 2022, doi:10.3390/bioengineering9100483_

Round 1
Reviewer 1 Report
Well structured and generally well performed work on stress distribution in implant restorations with inclined abutments.
Some criticisms are present:
A general sentence on implant stability issues in the abstract section should be added
-In the final section of the abstract, however, some CONSIDERATIONS on the possible clinical implications should be added-check that all keywords are pubmed MESH terms
-At the end of the introduction section, the null hypotheses of the study must be inserted, which will be refuted at the end of the discussion section
- it is necessary to specify directly which are the constraints provided to the geometry in the apical direction.
-How come these values of F were chosen applied to the geometry ?. The data are correct but it is necessary to take some bibliographic citations to support; in particular on the inclined load, does it refer to parafunctions?
-In the figures relating to the fesem the scales must all be the same to allow a better visualization of the results
-In the discussion section some considerations on the distribution of loads and forces between natural elements and systems should be added.
In this regard, I recommend that you insert the following scientific work in the reference section, which could be of help to the reader
Chieruzzi M, Pagano S, Cianetti S, Lombardo G, Kenny JM, Torre L. Effect of fiber posts, bone losses and fiber content on the biomechanical behavior of endodontically treated teeth: 3D-finite element analysis. Mater Sci Eng C Mater Biol Appl. 2017; 74: 334-346. doi: 10.1016 / j.msec.2016.12.022
.-A section on the limits of the study is missing
Reviewer 2 Report
01
There are some sentences in the text without reference to a previous study (or studies) in order to give evidence to their statements. Without references, these statements would be mere assumptions or allegations by the authors of the manuscript. Therefore, each of the following sentences need at least one reference to back up their statement:
“When two or more implants are placed, there is a high possibility of prostheses misconnection if the three-dimensionally misaligned angle exceeds 22o.”
“This phenomenon cannot be easily distinguished by cone-beam computed tomography (CBCT) or radiography, and when an incorrect tightening is made, constrained tightening may occur due to the tightening force of the abutment screw.”
“In such cases, stress concentration of the masticatory force occurs due to the misaligned angle, leading to the prosthesis, abutment, or fixture fracture.”
“Connecting the implant prostheses of the molars effectively disperses the occlusal force applied to them, efficiently solving the food-packing phenomenon caused by the loss of contact between the prostheses.”
“The implant prostheses are also connected if the bone quality of the area of the fixture placement is poor, the patient's occlusal force is strong, or if the diameter or length of the implant is narrow or relatively short, respectively.”
“Consequently, ISFDP misconnection is frequently observed in clinical cases.”
02
One of the authors of study works for the manufacturer of the implant components. The authors need to disclose any possible conflict of interest concerning this.
03
The following paragraphs consist mainly of a repetition of the results, without an actual discussion of the findings.
“The PVMS value of the fixture was lower in the AO group than in the BD group. This difference is probably due to the different thicknesses of the fixtures in both groups. (AO fixture was 0.1 mm thicker than the BD fixture). The highest stress was generated in the contact area of the abutment and the fixture. When a lateral load was applied, the stress was concentrated in the octagonal structure within the fixture and the part preventing the abutment from sinking; however, it was 54% of the yield stress (880 MPa). As a result, the PVMS value of the abutment was lower in the BD group than in the AO group.”
“Moreover, the PVMS value of the abutment screw in the BD group reinforced by in-309 creasing the diameter was lower than the PVMS value of the abutment screw in the AO group. Therefore, the risk of abutment and abutment screw fracture in the BD group was lower. In the AO group, the highest stress was generated in the connection area of the fixture and abutment screw. In particular, the stress was excessively concentrated around the abutment screw and abutment tightening. The PVMS value of the abutment in the AO group was 69% of the yield stress. Therefore, the fracture risk in the AO group was higher than in the BD group.”
For the abutment screw, the AO group presented the highest stress in the area in contact with the abutment, and the BD group had the highest stress in the area attached to the fixture. Additionally, the highest stress was generated in the fourth to fifth thread positions in the AO group, and the BD group had the highest stress in the fourth thread region. These differences in stress distribution are expected to affect the failure mode.”
04
Please identify and discuss the limitations of the study.
Round 2
Reviewer 2 Report
The manuscript now seems to be suitable for publication.